# Network Analysis of Biomarkers Associated with Occupational Exposure to Benzene and Malathion

**DOI:** 10.3390/ijms24119415

**Published:** 2023-05-28

**Authors:** Marcus Vinicius C. Santos, Arthur S. Feltrin, Isabele C. Costa-Amaral, Liliane R. Teixeira, Jamila A. Perini, David C. Martins, Ariane L. Larentis

**Affiliations:** 1Studies Center of Worker’s Health and Human Ecology (CESTEH), Sergio Arouca National School of Public Health (ENSP), Oswaldo Cruz Foundation (FIOCRUZ), Rio de Janeiro 21041-210, RJ, Brazil; 2Center for Mathematics, Computation and Cognition, Federal University of ABC, Santo André 09210-580, SP, Brazil; 3Research Laboratory of Pharmaceutical Sciences (LAPESF), State University of Rio de Janeiro (West Zone-UERJ-ZO), Rio de Janeiro 23070-200, RJ, Brazil

**Keywords:** network medicine, occupational health, malathion, benzene

## Abstract

Complex diseases are associated with the effects of multiple genes, proteins, and biological pathways. In this context, the tools of Network Medicine are compatible as a platform to systematically explore not only the molecular complexity of a specific disease but may also lead to the identification of disease modules and pathways. Such an approach enables us to gain a better understanding of how environmental chemical exposures affect the function of human cells, providing better perceptions about the mechanisms involved and helping to monitor/prevent exposure and disease to chemicals such as benzene and malathion. We selected differentially expressed genes for exposure to benzene and malathion. The construction of interaction networks was carried out using GeneMANIA and STRING. Topological properties were calculated using MCODE, BiNGO, and CentiScaPe, and a Benzene network composed of 114 genes and 2415 interactions was obtained. After topological analysis, five networks were identified. In these subnets, the most interconnected nodes were identified as: IL-8, KLF6, KLF4, JUN, SERTAD1, and MT1H. In the Malathion network, composed of 67 proteins and 134 interactions, HRAS and STAT3 were the most interconnected nodes. Path analysis, combined with various types of high-throughput data, reflects biological processes more clearly and comprehensively than analyses involving the evaluation of individual genes. We emphasize the central roles played by several important hub genes obtained by exposure to benzene and malathion.

## 1. Introduction

Complex diseases are associated with the effects of multiple genes, proteins, and biological pathways [1]. Therefore, in the case of a complex disease, one should not expect that a single genetic mutation can be identified as the cause. In fact, complex diseases or disorders (e.g., cancer, AIDS, and obesity) stem from dysfunctions of different biomolecular networks and not only their isolated components (e.g., genes, proteins, and metabolites) [2].

Biological networks are powerful resources for the discovery of genes and genetic modules that drive disease. Biomolecular networks include gene and transcription regulatory networks, protein-protein interaction networks, metabolic, signaling, and hybrid networks. With advances in high-throughput measurement techniques such as microarray, RNA-seq, ChIP-on-chip, and mass spectrometry, large-scale biological datasets have been continuously produced. Such data contain detailed information to understand the mechanism of molecular biological systems and have proven to be useful in the diagnosis, treatment, and design of drugs for complex diseases or disorders [3].

In this context, the Network Medicine [4] aims to use complex network analysis to find clusters (or modules) in a biological network that could be related to a phenotype of interest based on a few hypotheses, such as the hypothesis that a protein-protein interaction (PPI) network follows a scale-free (power law) degree distribution and that genes associated with the same (or similar) pathways are clustered closely in the PPI networks; therefore, they have a high probability of interacting with each other. This is a fundamental concept that can be used to combine and amplify signals from individual genes [5], genes with similar expression patterns [6], synthetic lethality [7], or chemical sensitivity [8], which often present similar functions. In addition, genes whose products interact physically [4,9] are part of the same complex [10], display similar three-dimensional structures [11], similar phylogenetic profiles [12], or have common protein domains [13]. Therefore, a biological pathway plays an important role in understanding the mechanisms of complex diseases, improving clinical treatment, and revealing drug targets and biomarkers [14]. Such an approach opens the possibility of a better understanding of how environmental chemical exposures affect the function of human cells, providing better insights into the mechanisms involved and assisting in the monitoring/prevention of exposure [15]. For instance, benzene is classified by the International Agency for Research on Cancer (IARC) as a carcinogen belonging to group I, i.e., carcinogenic to humans [16]. Exposure to benzene is associated with the occurrence of hematotoxicity, acute myeloid leukemia, and myelodysplastic syndromes [17,18,19,20,21].

Hematotoxicity effects are observed even at relatively low concentrations [22,23,24,25], although the hematopoietic toxicity mechanisms of action in benzene exposure remain unclear and are still under study, mainly by applying toxicogenomics techniques [26,27,28].

As is well known, benzene metabolism creates many reactive elements [16,29], and exposure to benzene and its metabolites can generate DNA mutations, chromosome insertions and/or deletions, DNA double-strand breaks, apoptosis, oxidative stress, and altered gene expression [28].

Another relevant environmental chemical exposure is related to malathion, which is a likely carcinogen due to the increased risk of cancer associated with its exposure [30,31]. Farm workers and their children, particularly, face an increased risk of developing leukemia and non-Hodgkin’s lymphoma due to their exposure to malathion [31,32,33]. In fact, studies have shown that malathion induces chromosomal and DNA damage in humans [34,35]. Thus, the IARC—a specialized cancer agency of the World Health Organization (WHO)—also classified malathion as ‘probably carcinogenic to humans’ (Group 2A). However, molecular changes caused by exposure to malathion have not been extensively explored to date, although neurological malignancies are prevalent in humans.

The biological processes and molecular functions underlying such exposures constitute complex systems [36], which cannot always be designated by a simplistic view such as assigning functions to individual genes, proteins, and other cell macromolecules [37]. In this context, several studies have presented methods to analyze and identify essential genes and proteins in a biological interaction network. Luo et al. [38] proposed a method to predict essential proteins in PPI networks based on local interaction density and protein complexes. Wang et al. [39] proposed a method to identify essential proteins by combining information about protein complexes and topological features of the PPI network. Hu et al. [40] address a method to be applied to weighted networks by considering the total strength of the interaction, the number of edges of the interaction, and the distribution of the total strength of the connection at the edge of the connection in the local domain. Cinaglia and Cannataro [41] and Dai et al. [42] address the use of static network alignment methods adapted to the dynamic context by performing network discrimination and providing other additional information. By reporting how various problems can be transferred from static networks to dynamic networks, taking into account temporal information. In addition, many of the key factors for measuring nodes in complex networks are based on graph theory to quantify the topological structure and attributes of each node, and comparisons of the centrality of each node are made by using different centrality calculation methods, such as degree center, median center, proximity center, and edge clustering coefficient center. Quantitative methods are also used to find the essential nodes in networks [43,44,45,46].

Considering these issues, this study adopted a systems biology approach to reconstruct networks of molecular interactions based on differentially expressed genes (DEG) related to benzene and malathion exposure, retrieved from literature text mining. Using some of the Network Medicine hypotheses, such as the disease module and the local hypotheses, our goal is to use network analysis to evaluate possible biological pathways associated with the response to benzene and malathion. Gene interactions were observed using the GeneMANIA software (version 3.5.2), while interactions between proteins were detected using the Search Tool for Retrieval of Interacting Genes (STRING) database (version 11.0). The Molecular Complex Detection (MCODE) software (version 2.0.2) was used for the characterization of clusters of different biological processes, and the Biological Networks Gene Ontology (BiNGO) software (version 3.0.5) was used for ontological gene characterization. The Cytoscape software (version 3.6.0) was used to calculate different network centrality metrics from these genes.

## 2. Results

To create, analyze, and select the hub genes related to exposure to benzene and malathion, a total of 96 human DEGs were selected through a bibliographic search for articles published between 2010 and 2014, dealing with quantitative microarray data for environmental and occupational exposure to benzene (Table 1). Additionally, 57 human DEGs were selected through a bibliographic search for articles published with quantitative microarray data for malathion exposure (Table 2). Figure 1 shows the entire experimental design, and a list of all genes selected for benzene and malathion bibliographic searches, in addition to the list of genes added outside the list in the network creation stage by STRING and GeneMANIA, is present in Appendix A.

### 2.1. Interaction Network

The analysis of biological interactions by GeneMANIA revealed a network comprising 114 genes and 2415 interactions (Figure 2A) and a PPI network comprising 67 nodes and 134 edges by STRING. 

The gene-gene network presented a predominance of co-expression interactions (Figure 2B), followed by physical interactions (Figure 2C), molecular pathways (Figure 2D), predicted (Figure 2E), co-localization (Figure 2F), genetic interactions (Figure 2G), and shared protein domains (Figure 2H), respectively.

### 2.2. Cluster Analysis

The MCODE software (version 2.0.2) classified and separated molecular aggregates into clusters. The cluster analysis identified five sub-networks (Figure 3), ranked based on node connectivity level (score).

### 2.3. Centiscape Analysis

In order to search for network biomarkers, the structure and manner of information flows (connectivity) along the networks were evaluated (Figure 4 and Figure 5 and Table 3).

For Benzene network exposure, in the 1st cluster, *IL8* was identified as the bottleneck node with degree = 34.12903, betweenness = 3.03225, and eigenvector = 0.18904. In the 2nd cluster, *KLF4*, *KLF6*, and *JUN* were identified as the bottleneck nodes with degree = 12.625, betweenness = 5.25, and eigenvector = 0.23.

In the 3rd cluster, *SERTAD1* and *MT1H* were identified as bottleneck nodes, with degree = 8.25, betweenness = 9.125, and eigenvector = 0.23. The 4th and 5th clusters are poorly interconnected, with central nodes in *DEPDC1* and *ARHGAP19* (degree = 3; betweenness = 0.5, and eigenvector = 0.49) and *SLC38A2* (degree = 3; betweenness = 6.6, and eigenvector = 0.5), respectively (Figure 4).

In the PPI network, *HRAS* and *STAT3* were identified as the bottleneck nodes with degree = 5.5, betweenness = 52.95, and eigenvector = 0.28 (Figure 5).

### 2.4. GO Overrepresentation Analysis (BiNGO)

Of total 390 ontologies (Appendix A) regulated by DEGs in the Benzene network, the GO (Gene Ontology) enrichment analysis identified up-frequency of Overrepresentation (adjusted *p* value < 0.05) in Biological Processes for “biological regulation”, “ regulation of the biological process”, and “regulation of the cellular process”, while down-frequency of Overrepresentation in Biological Processes related to the “positive regulation of nitric oxide biosynthetic process”, “regulation of vascular endothelial growth factor production”, “cytokine-mediated signaling pathway”, “leukocyte migration”, “leukocyte chemotaxis”, “cell chemotaxis”, and the most significant terms were “response to chemical stimulus”, indicating that the regulation of factors related to cellular locomotion, mainly leukocytes, plays a central role on the relationship between benzene and the development of leukemia (Figure 6).

Of total 314 ontologies (Appendix A) regulated by DEGs in the Malathion network, the GO enrichment analysis identified up-frequency of Overrepresentation (adjusted *p* value < 0.05) in Biological Processes for “regulation of cellular process”, “regulation of biological process”, and “biological regulation”, while down-frequency of Overrepresentation in Biological Processes related to “negative regulation of apoptosis”, “negative regulation of cell death”, “negative regulation of programmed cell death”, “transmembrane receptor protein tyrosine kinase signaling pathway”, and the most significant terms were “positive regulation of biological process”, indicating more general biological functions and cell death regulation (Figure 7).

## 3. Discussion

STRING and GeneMANIA are two well-known and simple-to-use web tools, with no need for the user to have advanced knowledge in programming to use them. When compared to other tools such as the Alignment-Based Network Construction Algorithm (ANCA) [53], their simple interface, web access, and the possibility that they can produce a list of associated genes from a query based on several biological associations make these tools more attractive for users.

These two tools provide the same service for four biological associations (physical interaction, genetic interaction, co-expression, and co-citation), but other associations are unique to a particular tool. STRING, for example, has three unique biological associations (gene fusion evidence, co-occurrence, and pathway evidence), as well as GeneMANIA (co-inheritance, colocalization, and shared protein domains).

STRING provides a limited variety of association network data to be used for a given query. In contrast, GeneMANIA generates customized results for the consulted genes and the data sources selected by the user. However, GeneMANIA needs extensive literature to filter the results (e.g., the co-expression data, which is a database of cancer genomes), while STRING may be more suitable for cases where not so much data is available. This is precisely why STRING was adopted for malathion DEGs, since not many data exist for humans (according to IARC classification [8], malathion is considered 2A—“probably carcinogenic to humans”).

Among the advantages observed when using GeneMANIA is the potential to expand the search for related genes, which in this study represented 96 genes obtained from the literature, for a final network comprising 114 genes (Appendix A). Some of the added genes also show altered expression upon benzene exposure (such as *JUN*, *KLF6*, *KFP36,* and *BCL2A1*) [54,55,56]. However, one of the limitations of using this software (version 3.5.2) is its inability to identify gene synonyms, as GeneMANIA did not identify *TRA@*, *AD022,* and its synonyms as valid names, as well as p15, although the official name *CDKN2B* was identified. In addition to its ability to detect and compensate for data redundancy, the GeneMANIA Prediction Server also displays an advantage due to the predictive accuracy and propagation of its algorithm [57,58].

With both approaches demonstrating potential in their use to identify hub genes with biological plausibility in identifying alteration/characterization of exposure and possible use as a target for cancer diagnosis, prognosis, and treatment. For example, the hub genes identified in this study are:

Benzene exposure:

*IL-8* is a chemokine related to the promotion of chemotaxis and neutrophil degranulation. This chemokine activates multiple intracellular signaling pathways and is a significant regulatory factor within tumor microenvironments [54]. Increased *IL-8* expression is present in several cell types in the presence of benzene metabolites [55,59,60]; Gillis et al. [58] have demonstrated that human peripheral blood mononuclear cells (PBMCs) induce the production of cytokines in the presence of benzene metabolites, such as IL-8, at levels from 10 to several thousand-fold, resulting in increased cytokine levels in the medium, while the effects of benzene metabolites on the secretion of soluble cytokines are varied. For example, reductions in IL-8 production dependent on hydroquinone and catechol concentrations but not on benzenetriol and benzoquinone have been observed. This demonstrates the relevance of network visualization as a whole, not just the possibility of direct *IL-8* changes, where the most specific Biological Process of the Benzene network are related to cytokine-mediated signaling pathways.

Krüppel-like factors (KLFs) are highly conserved zinc-finger proteins that regulate cellular transcription machinery and regulate a wide range of cellular functions, including cell proliferation, apoptosis, differentiation, and neoplastic transformation, by binding to GC-rich promoter regions [51,56]. *KLF6* and the proto-oncogene *JUN* are significant in differentiation and cell death, hematopoiesis, and cell survival. Regarding functions, all categories linked to DNA structure and transcription were present (in the case of benzene exposure and the identification of possible biomarkers, this becomes important) [51]. Unlike the *JUN* gene, which is known to regulate myeloid differentiation [61], the KLF6 gene is a known tumor suppressor for prostate [62], colorectal [63], lung [64], ovary [65], gliomas [66], head and neck [67], and hepatocellular cancer [68]. However, the *KLF6* gene inhibits *JUN*-dependent transcription, which leads to an antagonistic effect on cell proliferation induced by the *JUN* gene [69]. This demonstrates both the non-specificity of benzene-inducing changes and the importance of a biological interaction approach.

*SERTAD1* antagonizes the function of the inhibitor of apoptosis-stimulating protein p53 (iASPP), preventing its entry into the nucleus to interact with p53 in leukemic cells when iASPP is in its overproduction stage [70]. Evidence associating increased *SERTAD1* expression with the presence of benzene and its metabolite benzoquinone has been reported in the literature [51].

The *HRAS* protein is a small GTPase that cycles between inactive and active conformations. In the active state, *HRAS* binds to guanosine triphosphate (GTP) and possesses an intrinsic enzymatic activity that cleaves the terminal phosphate of this nucleotide, converting it to GDP. Upon conversion of GTP to guanosine diphosphate (GDP), HRAS is made inactive [70]. *STAT3* is a member of a family of cytoplasmic proteins that participate in cellular responses to cytokines and growth factors as transcription factors. Signal transducers and activators of transcription (STATs) are transcription factors that transmit signals from the extracellular surface of cells to the nucleus. *STAT3* is phosphorylated and activated in response to interleukin-6, contributing to an increased expression of genes activated in the liver during the acute phase response to inflammation [71].

Malathion network context:

Dysregulation of *HRAS* and *STAT3* pathways is frequently observed in several cancers [72,73,74]. Then altered HRAS protein is permanently activated within the cell. This overactive protein directs the cell to grow and divide in the absence of outside signals, leading to uncontrolled cell division and the formation of a tumor [2]. *STAT3* regulates basic biological processes essential to tumorigenesis, including cell-cycle progression, apoptosis, tumor angiogenesis, invasion and metastasis, and tumor-cell evasion of the immune system [75].

GO analysis and pathway enrichment provided further insights about Biological Processes related to Benzene and Malathion networks. The interrelationship between pathways and genes can be observed as the most frequent pathways represent more general processes and the least frequent pathways represent more specific gene processes. Such Benzene network functions are associated with leukemia mechanisms, and several of these functions and processes are mediated by *IL1A* and *PTGS2*, which play a central role in the characterization of gene expression associated with benzene exposure [47]. *IL1A* exhibits a single nucleotide polymorphism (SNP), which increases the expression of its mRNA and is inversely associated with granulocyte counts in benzene-exposed individuals [56], while *PTGS2* overexpression frequently occurs in pre-malignant and malignant neoplasms, including hematological malignancies [76]. The Malathion network is more associated with cellular death regulation mechanisms.

Therefore, to approach a complex disease study, a useful clue is provided by the fact that genes, gene products, and small molecules interact with each other to form a complex interaction network. Thus, an alteration in one gene might propagate through interactions, possibly affecting other genes in the network. The fact that one can observe similar disease phenotypes despite different genetic causes suggests that these different causes are not unrelated but rather jointly contribute to dysregulating the same component of the cellular system [1]. However, it is worth noting that exposure to low concentrations of environmental and occupational carcinogens does not eliminate the risk, considering that there is no safe exposure limit to carcinogenic substances such as benzene and malathion [77,78,79].

Moreover, one of the limitations of this study is that the representation of these complex networks of undirected biological interactions represents only a large number of possible pathways and interactions without taking into account the direction and dynamics of these processes. Although this approach shows potential to identify hub genes with biological plausibility in identifying alteration/characterization of exposure and possible use as a target for cancer diagnosis, prognosis, and treatment. In the limiting case, undirected network analysis assumes that all biological interactions and pathways represent processes occurring at the same rate, and then the observed clustering may lead to unrealistic conclusions. For example, interaction dynamics (chemical reaction kinetics) may promote some pathways while inhibiting others, just as the expression of some genes inhibits that of others [70,80]. For this reason, additional studies are recommended, mainly those involving directed networks. Once the directions in which information flows in the network are known, it can lead to a better identification of potential markers for diagnosis, treatment, and prevention.

## 4. Materials and Methods

### 4.1. Data Collection

Text mining was carried out using the bibliographic searches of NCBI (PubMed), Agilent Literature Search, Gene Expression Omnibus (GEO), Reactome, ArrayExpress, and Medline, involving two queries: one with the keywords “benzene gene expression”, “benzene microarray”, “benzene expression”, and “benzene poisoning;”, and another with the keywords “malathion gene expression”, “malathion microarray”, “malathion expression”, and “malathion poisoning”.

### 4.2. Construction of the Protein-Protein Interaction (PPI) Network

The Search Tool for Retrieval of Interacting Genes (STRING 11.0; http://string-db.org/ accessed on 22 February 2022) was used to evaluate the PPI of DEGs related to malathion exposure. The PPI network was derived from proven experimental statistics such as automated text mining of scientific literature, experimental data, available signaling pathways, the PPI database, systematic coexpression, phylogenetic co-occurrence, observation of neighboring genomes, and genetic fusion events. We only extracted the edges with a minimum confidence score of 0.7 (high confidence) and the maximum number of interactors at the first shell with no more than five interactors.

The visualization of network and module analysis was carried out with the software Cytoscape (version 3.6.0; http://www.cytoscape.org/ accessed on 15 February 2018).

### 4.3. Extended Interaction Network Analysis

The obtained gene list corresponding to benzene exposure was used for a functional and gene ontology (GO) analysis using GeneMANIA (version 3.5.2; http://genemania.org/ accessed on 17 February 2018), a tool used for predicting gene function that can be implemented as a plug-in for the Cytoscape networks visualization software (version 3.6.0; http://www.cytoscape.org/ accessed on 15 February 2018) [81].

The biological data sets searched by GeneMANIA, the characteristics considered for interaction formation, and the data source for the network creation are shown in Table 4.

For the analysis, we applied the default setting of 20 genes, which present the highest number of interactions. The software (version 3.5.2) standard was maintained regarding the advanced settings for physical, genetic, and interrelated paths, only modifying the co-expression data for articles on genomic leukemia and other hematological diseases.

### 4.4. Identification of Molecular Complexes

Highly interconnected, or dense, regions may represent molecular complexes. The Molecular Complex Detection (MCODE) algorithm, an automated method for finding clusters (highly interconnected regions), was used to identify sets of molecules that strongly interact with each other in the Benzene network. In addition, MCODE standard parameters were maintained (degree cutoff = 2, node score cutoff = 0.2, K-core = 2, and maximum depth = 100).

### 4.5. GO Category Representation

The biological network gene ontology (BiNGO) tool was used to perform a GO functional enrichment analysis. The hypergeometric distribution was used for the biological processes and molecular functions of functional overrepresentation categories. Only the over-represented GO categories were considered significant (adjusted *p* value < 0.05).

### 4.6. Centrality Analysis

Centralities are parameters that identify nodes with relevant positions in the global network architecture. The CentiScaPe 2.2 plug-in was used for clusters displaying an overrepresentation of GO categories of interest.

CentiScaPe calculates specific centrality parameters, describes the network topology, and assists in identifying the most important nodes in a complex network [82]. According to the connectivity of each node degree, betweenness, and eigenvector, the arithmetic mean of each centrality parameter was defined to obtain the most connected nodes.

Degree centrality indicates the number of adjacent nodes that are connected to a unique node. If the degree of centrality of a given node is much larger than the average degree of centrality of the network, such a node is classified as a hub. Hubs in interactive networks tend to be essential since their exclusion reduces the connectivity of the global network; consequently, they also represent greater biological relevance [4,83]. Betweenness, in turn, is defined by the number of shortest paths between all pairs of nodes that pass through a node of interest [82,84]. The nodes with the highest betweenness score (the ‘bottleneck’) control most of the information flow in the network, as they present the largest number of shortest paths (between other nodes in the network) passing through. As such, bottleneck genes (nodes in the interactome ranked by betweenness centrality) are related to regulatory functions, representing critical points in the PPI network [85]. The eigenvector defines the node “prestige” of a network, i.e., the eigenvector centrality of a node in a network is large if this node is connected to many central and highly connected nodes [86].

## 5. Conclusions

Complex diseases, especially cancer, are extremely harmful to human health. Therefore, the identification of biomarkers is key to dealing with complex disease studies. Pathway analysis, combined with multiple types of high-throughput data, reflects biological processes more clearly and comprehensively than analyses involving the assessment of individual genes. For this reason, pathway-based complex disease analysis approaches have become a hot research topic.

In this study, we began with the systematic analysis of associated genes using text mining, followed by the identification of essential genes and pathways by functional annotation. We emphasize the central roles performed by several important hub genes, such as proteins from the *IL8*, *KLF*, *JUN*, and *SERTAD1* families obtained by exposure to benzene, while *HRAS* and *STAT3* are hubs for exposure to malathion.

Although the network results corroborate the literature about hub genes plausibility as potential biomarkers, additional validation studies are required to confirm this hypothesis. Given that systems biology approaches for predicting and characterizing hub genes are recent [82,83,84,85], the way in which data collection and network creation are conducted can vary according to the purpose of the study.

## Figures and Tables

**Figure 1 ijms-24-09415-f001:**
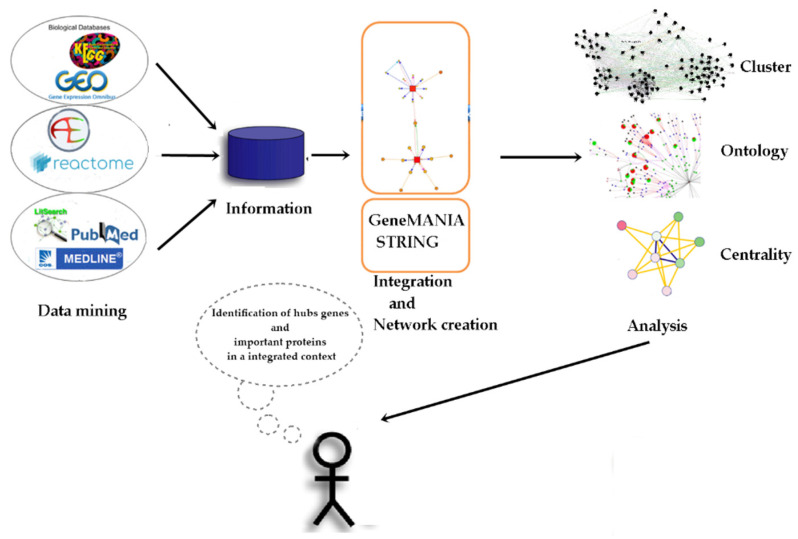
Main stages of the experimental project.

**Figure 2 ijms-24-09415-f002:**
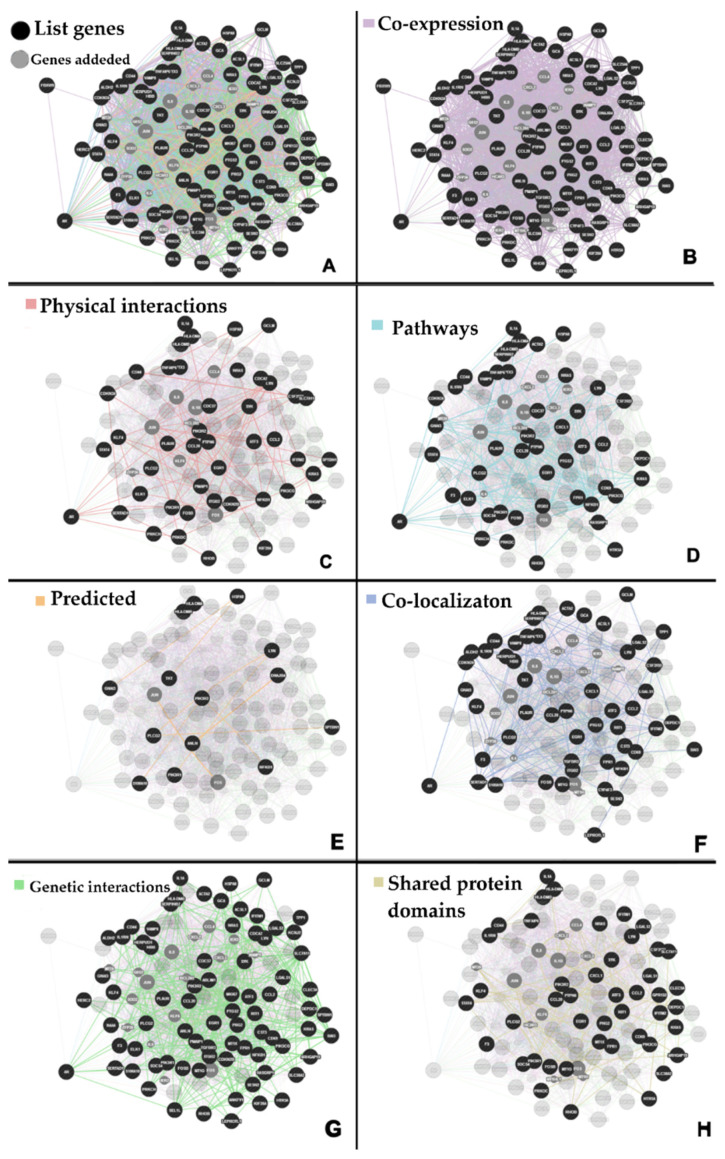
Benzene biological interaction network. (**A**) Interaction network generated using GeneMania with 114 genes comprising 2415 interactions; (**B**) co-expression interactions; (**C**) physical interactions; (**D**) pathways; (**E**) predicted; (**F**) co-localizaton; (**G**) genetic interactions; and (**H**) shared protein domains.

**Figure 3 ijms-24-09415-f003:**
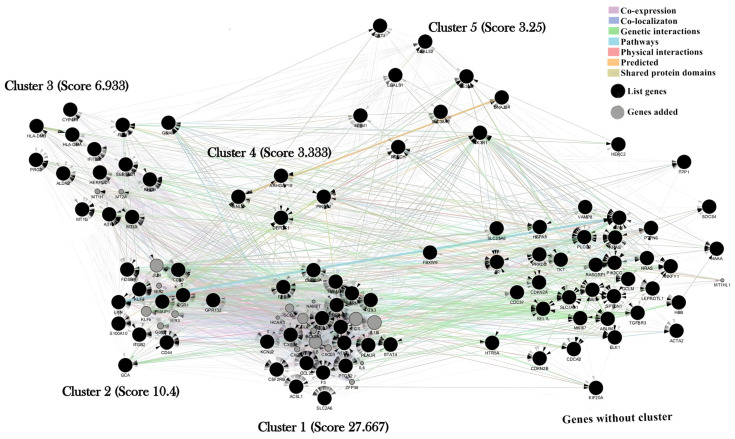
Overview of the five clusters obtained from the benzene gene-gene network using the MCODE software. Interaction network generated using GeneMANIA with 114 nodes (8 nodes expanded from a 96-gene list) and 2415 edges. All nodes are interconnected in a unique connected component, but 38 nodes do not belong to any cluster: Cluster 1: score = 27.667; Cluster 2: score = 10.4; Cluster 3: score = 6.933; Cluster 4: score = 3.333; and Cluster 5: score = 3.25. Nodes may represent biological elements, while the edges describe the nature of their relationships (co-expression; physical; pathways; predicted; co-localizaton; Genetic interactions; and shared protein domains).

**Figure 4 ijms-24-09415-f004:**
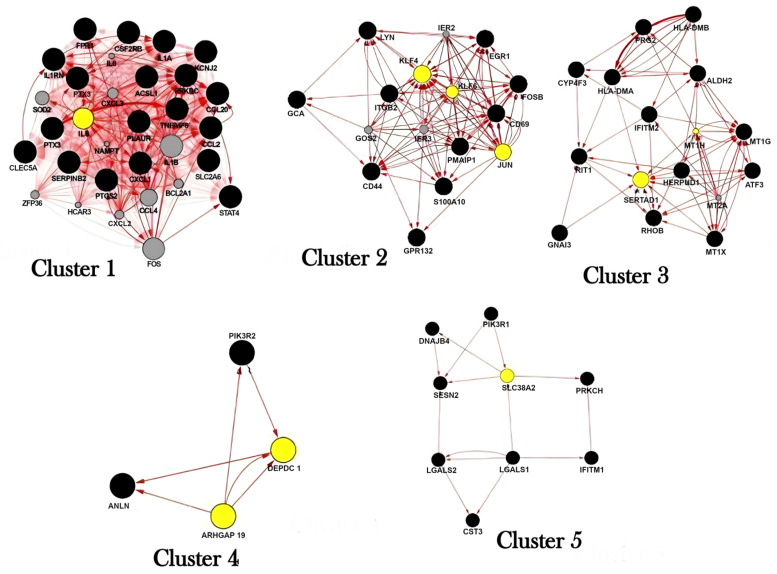
Most relevant nodes (in yellow) for the subnetworks predicted by the MCODE analysis in the benzene gene-gene network. Centrality is calculated by node degree, betweenness, and eigenvector. Cluster 1, IL8 was identified as the bottle-neck node; Cluster 2, KLF4, KLF6, and JUN were identified as the bottleneck nodes; Cluster 3, SERTAD1 and MT1H were identified as bottleneck nodes and Clusters 5 DEPDC1 and ARHGAP19 were identified as bottleneck nodes.

**Figure 5 ijms-24-09415-f005:**
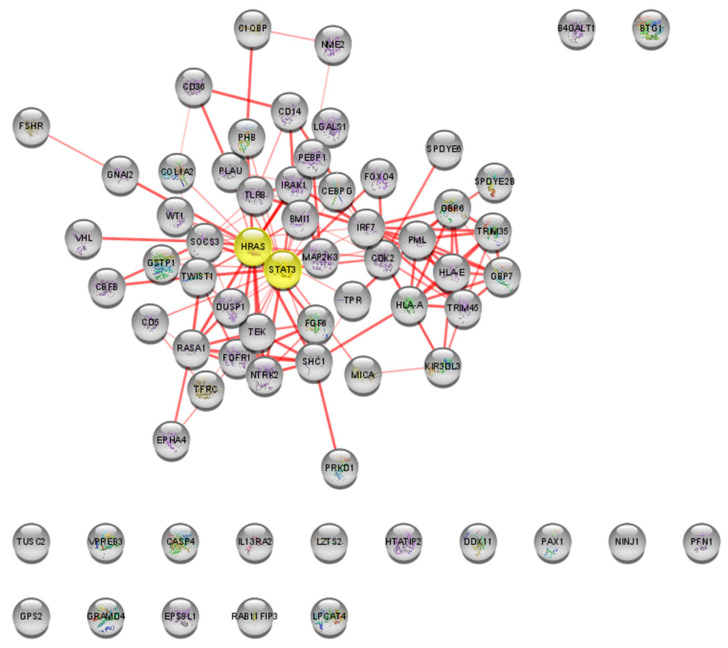
Most relevant nodes (in yellow) for Malathion PPI network analysis in STRING. Centrality is calculated by node degree, betweenness, and eigenvector. Interaction network generated using STRING with 67 nodes (10 nodes expanded from a 57-gene list) and 134 edges. Fifty nodes are interconnected in the main connected component, while 17 nodes are isolated.

**Figure 6 ijms-24-09415-f006:**
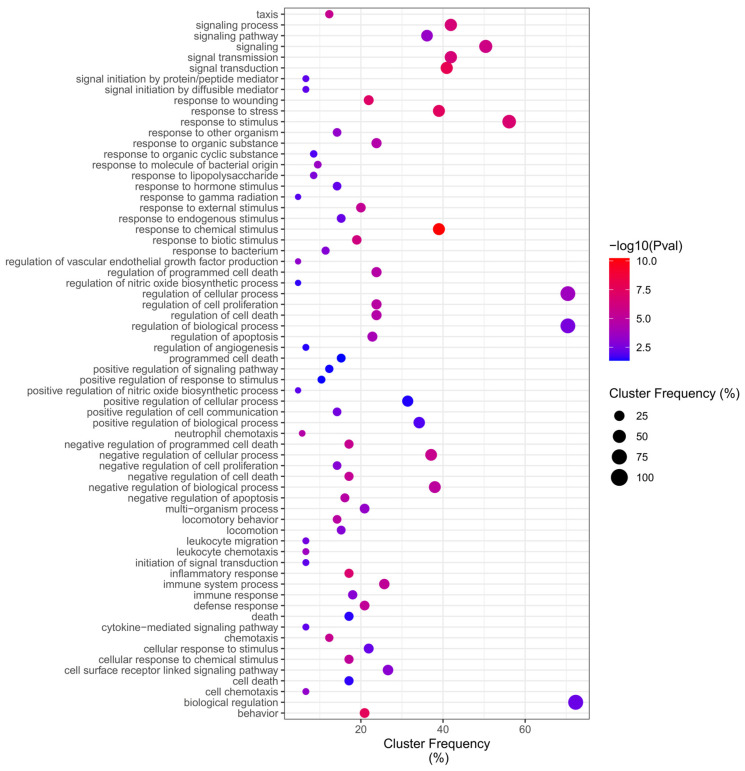
Benzene Biological Process Overrepresentation Analysis (BiNGO).

**Figure 7 ijms-24-09415-f007:**
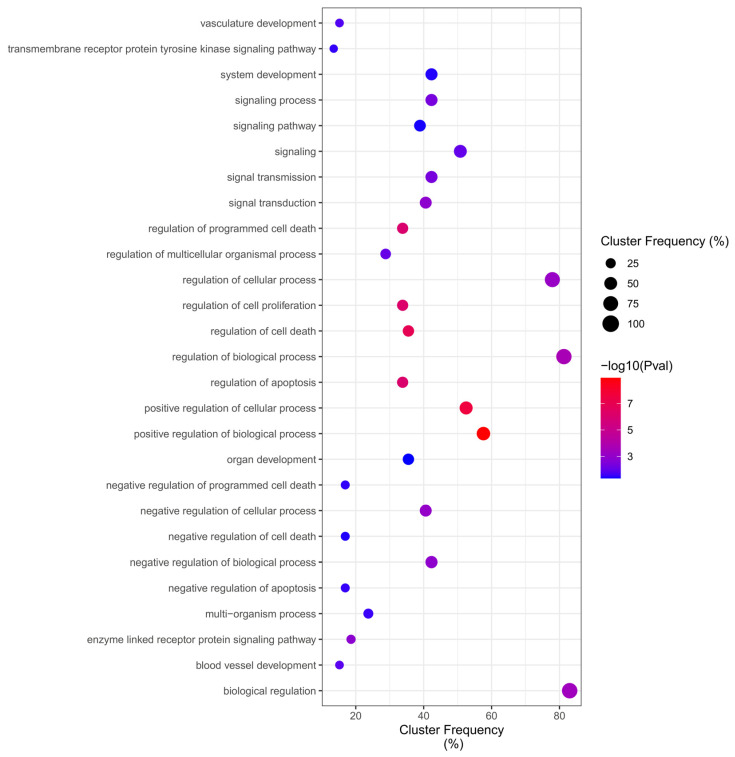
Malathion Biological Process Overrepresentation Analysis (BiNGO).

**Table 1 ijms-24-09415-t001:** Benzene list of the most important studies obtained herein and their selected genes.

Reference	Method	Exposure	Controls	DEGs
McHale et al. [47]	Microarray	83 cases of benzene exposure ranging from <<1 to >10 ppm.	42	16 genes with high expression (*SERPINB2*, *TNFAIP6*, *IL1A*, *KCNJ2*, *PTX3*, *F3*, *CD44*, *CCL20*, *ACSL1*, *PTGS2 CLEC5A*, *IL1RN*, *PRG2*, *SLC2A6 GPR132*, and *PLAUR*).
Bi et al. [48]	cDNA microarray	7 women were diagnosed with benzene poisoning.	7	Top 40 genes with altered expression (*PTGS2*, *BAI3*, *GCL*, *CYP4F3*, *MY047*, *TRA@*, *AD022*, *PRKCH*, *RASGRP1*, *FPR1*, *TGFBR3*, *GRO1*, *SEL1L*, *CSF2RB*, *IFITM1*, *STAT4*, *IFITM2*, *ABLIM*, *KIAA1382*, *SPTBN1*, *HBB*, *PRKDC*, *S100A10*, *ITGB2*, *TKT*, *VAMP8*, *FOSB*, *ASAHL*, *CDC37*, *SLC25A6*, *CLN2*, *ACTA2*, *CST3*, *HLA-DMB*, *ALDH2*, *LGALS2*, *LGALS1*, *ARHB*, *KLF4*, *and ATF3*).
Xing et al. [49]	Microarray (RTPCR)	11	People in the same sector with no symptoms of benzene poisoning.	Decrease in the expression of p15 (*CDKN2B*) and p16 (*CDKN2A*).
Sarma et al. [50]	Microarray	Culture of HL-60 cells treated with IC50 concentrations of benzene, hydroquinone, and benzoquinone.	Culture of HL-60 cells treated with dimethyl sulfoxide.	Alteration in expression of 27 genes (*CCL2*, *EGR1*, *GCLM*, *PMAIP1*, *SESN2*, *CD69*, *HERPUD1*, *HSPA8*, *RIT1*, *SERTAD1*, *SLC38A2*, *SLC7A11*, *DNAJB4*, *ANKFY1*, *ANLN*, *AR*, *ARHGAP19*, *CDCA2*, *DEPDC1*, *ELK1*, *FBXW9*, *HERC2*, *HTR5A*, *KIF20A*, *MKI67*, *MT1G*, and *MT1X*).
Gao et al. [51]	cDNA microrray	4 people were diagnosed with benzene poisoning, and 3 people from the same factory were exposed but had no symptoms.	3	Top 14 significant genes with altered expression (*PIK3R1*, *PIK3CG*, *PIK3R2*, *GNAI3*, *SYK*, *PTPN6*, *KRAS*, *NRAS*, *PLCG2*, *NFKB1*, *LYN*, *SOCS4*, *HLA-DMA*, and *HLA-DMB*).

**Table 2 ijms-24-09415-t002:** Malathion list of the most important studies obtained herein and their selected genes.

Reference	Method	Exposure	Controls	DEGs
Anjitha et al. [52]	Microarray	Culture of human lymphocytes treated with three concentrations of malathion (50, 100, and 150 μg/mL).	Culture of human lymphocytes treated with DMSO (1%).	57 DEGs (*B4GALT1*, *BMI1*, *BTG1*, *C1QBP*, *CASP4*, *CBFB*, *CD14*, *CD5*, *CD36*, *CDK2*, *CEBPG*, *COL1A2*, *DDX11*, *DUSP1*, *EPHA4*, *EPS8L1*, *FGF6*, *FGFR1*, *FOXO4*, *FSHR*, *GNAI2*, *GPS2*, *GRAMD4*, *GSTP1*, *HLA-A*, *HLA-E*, *HTATIP2*, *IL13RA2*, *IRAK1*, *LGALS1*, *LPCAT4*, *LZTS2*, *MAP2K3*, *MICA*, *NINJ1*, *NME2*, *NTRK2*, *PAX1*, *PEBP1*, *PFN1*, *PHB*, *PLAU*, *PML*, *PRKD1*, *RAB11FIP3*, *SHC1*, *SOCS3*, *TEK*, *TFRC*, *TLR8*, *TPR*, *TRIM35*, *TUSC2*, *TWIST1*, *VHL*, *VPREB3*, and *WT1*)

**Table 3 ijms-24-09415-t003:** Most relevant nodes for Benzene and Malathion networks.

GeneMania Benzene Network		
Symbol	Name	HGNC-ID
KLF4	KLF transcription factor 4	6348
SERTAD1	SERTA domain containing 1	17932
IL8	Interleukin 8	6025
JUN	Jun proto-oncogene, AP-1 transcription factor subunit	6204
KLF6	KLF transcription factor 6	2235
MT1H	Metallothionein 1H	7400
String Malathion		
Symbol	Name	HGNC-ID
HRAS	Hras proto-oncogene, GTPase	5173
STAT3	Signal transducer and activator of transcription 3	11364

**Table 4 ijms-24-09415-t004:** List of interaction network categories demonstrating the information for the creation of the biological network interactions and the data source researched by GeneMANIA.

Network Categories	Datasets	Network Information	Data Source
Co-expression:	Gene expression	Two genes are linked if their expression levels are similar across conditions in a gene expression study.	Most of these data are collected from the Gene Expression Omnibus (GEO).
Physical Interaction:	Protein-protein interaction	Two gene products are linked if they were found to interact in a protein-protein interaction study.	These data are collected from primary studies found in protein interaction databases, including BioGRID and PathwayCommons.
Genetic interaction:	Genetic interaction	Two genes are functionally associated if the effects of perturbing one gene are found to be modified by perturbations to a second gene.	These data are collected from primary studies and BioGRID.
Shared protein domains:	Protein domain	Two gene products are linked if they have the same protein domain.	These data are collected from domain databases such as InterPro, SMART, and Pfam.
Co-localization:	Genes expressed in the same tissue or proteins found in the same location.	Two genes are linked if they are both expressed in the same tissue or if their gene products are both identified in the same cellular location.	
Pathway:	Pathway	Two gene products are linked if they participate in the same reaction within a pathway.	These data are collected from various sources, such as Reactome and BioCyc, via PathwayCommons.
Predicted:	Predicted functional relationships between genes, often protein interactions.	For instance, two proteins are predicted to interact if their orthologs are known to interact in another organism. In these cases, network names describe the original data source of experimentally measured interactions and the organism from which the interactions were mapped from (e.g. a mouse network predicted from a human network).	A major source of predicted data is mapping known functional relationships to another organism via orthology. These data are collected from various sources, such as BioGRID and I2D orthology.

Source: GeneMANIA network categories (http://pages.genemania.org/help/ accessed on 28 February 2023).

## Data Availability

Data are contained within the article or Appendix A. The data that support the findings of our study are publicly available datasets from the GEO (https://www.ncbi.nlm.nih.gov/geo/), GeneMANIA (https://genemania.org/), and STRING (https://string-db.org/) databases, accessed on 22 February 2022.

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
