# Peer review of "Network Analysis of Biomarkers Associated with Occupational Exposure to Benzene and Malathion"

_ijms, 2023, doi:10.3390/ijms24119415_

Round 1

Reviewer 1 Report

This interesting paper reports on the analysis of molecular interactions networks based on differentially expressed genes (DEG) related to benzene and malathion exposure. The Authors have evaluated the possible gene interactions using GeneMANIA software as well as proteins interactions using STRING database. The characterization of clusters of different biological processes such as co-expression, co-localizaton, genetic interactions was performed using theoretical clustering algorithm, Molecular Complex Detection (MCODE). The biological network gene ontology (BiNGO) software was used for ontological gene characterization. The network topology, calculation of specific centrality parameters and identifying the most important nodes in a complex network was described by CentiScaPe bioinformatic tool. I recommend the paper for publication after minor revision addressing the issues listed below.

1.         The graph theory applied in this work enabled Authors to identify most important pathways and genes involved in disease/damage caused by occupational exposure to benzene and malathion. The Authors indicate that this is not a simple analysis of the effect of a single gene mutation or a damaged protein, but a complex process involving multiple genes, proteins, metabolites, etc. However, the presentation of these complex networks of biological interactions occurring in response to the toxic chemicals represents only a vast number of possible pathways and interactions without taking into account the dynamics of these processes and concentrations of species involved. In reality, the interaction dynamics (chemical reaction kinetics) may promote some pathways while hinder others. Therefore, the conditions for the Network Analysis performed should be clearly stated. In the limiting case, the Network Analysis may assume that all biological interactions and pathways represent processes occurring with the same rate and then, the observed clustering may lead to nonrealistic conclusions. The effect of the reaction kinetics for biological systems, such as the DNA strand replacement process, see: Stobiecka M and Chalupa A, DNA Strand Replacement Mechanism in Molecular Beacons Encoded for the Detection of Cancer Biomarkers. J Phys Chem B. 2016, 120, 4782-4790. This relevant literature reference should be cited.

2.         The numbers in text should be presented with the same (or close) number of significant digits.

3.         In Figures 1-2 and 3-6, markings of the cluster number, relevant nodes, pathways, etc. are illegible. Please increase the font size and resolution of pictures.

4.         The explanation of abbreviations should be provided at their first use, e.g., STRING, MCODE, BiNGO, GTP, GDP. For instance, the abbreviation “PPI” is used from the beginning of the article and not defined until page 11 line 314.

5.         There are some English corrections needed:

-          Abstract, line 16: “leading to” – replace with “but may also lead to”.

-          Abstract, line 17: “Such approach leads to the possibility of better understanding on how environmental chemical exposures …” – better would be: “Such an approach enables to gain better understanding of how the environmental chemical exposures …”.

-          Abstract, line 25: “In these subnets, IL-8, KLF6, KLF4, JUN, SERTAD1 and MT1H were the most interconnected nodes.” – smoother to read would be: “In these subnets, the most interconnected nodes were identified as: IL-8, KLF6, KLF4, JUN, SERTAD1 and MT1H.”

-          Page 2 line 82: “not very explored” – change to “not extensively explored”;

-          Page 2 line 93: “pathways associated with benzene and malathion response” – change to “pathways associated with the response to benzene and malathion”.

Author Response

Please see the attachment for response of the reviewer’s comments point-by-point.

We would like to thank the reviewer for evaluating our manuscript. All changer are in red and we have tried to address all concerns properly and we believe that our article has improved considerably.

Reviewer 2 Report

In this paper, the authors presented a systematic analysis of associated genes by using text mining, followed by the functional annotation for the identification of genes and pathways of interest. Tests reported proactive results that corroborate the literature.

Related works are missing, some generic information is only present in the introduction. I suggest reporting similar studies by presenting the methodologies used by these one, related to your study, in order to highlight the strengths (and/or weaknesses) of your study in relation to the literature. In my opinion, related works are essential to the reader.

Background is well-presented, however, I suggest including some notion about the networks, e.g., by mentioning a survey or other bioinformatics works focused on network analysis, such as: DOI: 10.1007/s13721-022-00383-1 (survey related to static and dynamic networks, focused on bioinformatics and biological data), DOI: 10.1145/3307339.3342152 (patterns and network evolution), DOI: 10.1109/BIBM.2018.8621193 (data-integration and disease-gene association).

Methodology is exhaustive; some concepts could be better exploded to provide more details. For instance, how did you create the network? …you generated an edge list, the data was in a format useful for the purpose, you identified the nodes and iteratively identified the edges.

Results and Discussion are well-described and well-explained. However, a comparative study between your results and ones obtained by another well-known software tool is suggested. In your manuscript, you did not report information about why you have chosen GeneMANIA, and no other ones like it.

I suggest deep checking the manuscript, carefully and thoroughly; to give an example, Table 3 reports “Title 3” “Title 4” as header of two columns (I think you missed overriding the template).

Majors:

- Related works are missing.

- Manuscript needs to be checked (e.g., table 3).

- Comparative studies with other ones or similar software tools could be included (alternatively, you have to motivate the choices related to your methods).

Minors:

- The manuscript contains typos and grammar mistakes.

- Abstract could be well-formatted, by adding carriage returns at the beginning of the subsection (if possible).

Author Response

(The authors gave the same response as above.)

Reviewer 3 Report

The article is clear but the discussion needs improvement. All the data is shared, and overall the article structure is good. There are few suggestions below:

Introduction - required some background information about benzene and malathion affecting human cells’ function and signaling mechanisms.

“A total of 96 Human DEGs were selected through an NCBI bibliographic search for 101 articles published between 2010-2014” - Is there no DEGs identified after 2014, why up to 2014? Please update with the recent studies.

“Additionally, 57 Human 103 DEGs were selected through bibliographic search for articles published with quantitative 104 microarray data for malathion exposure” – DEGs extracted only from (Anjitha et al. 2020) – Please check the other studies.

"The analysis of biological interactions by GeneMANIA revealed a network 114 comprising 114 genes and 2415 interactions (Figure 1A)" - How this Interactions incorporated into your findings or results? - No details explanation in the manuscript just mentioned the GeneMANIA output.

In the Discussion, you presented more about the well-known web tools and their limitations (firs 3 paragraph). I think you need to more focus on your major findings. For example, discuss the biological associations and how they lead to play a role in disease progression.

In Discussion – “And Malathion network context: 276 Dysregulation of HRAS and STAT3 pathways is frequently observed in several 277 cancers [63–64]” – There is no background of cancers and Malathion-related genes to involved in disease progression.

“With both approaches demonstrating potential in their use to identify hub genes biological plausibility in identifying alteration/characterization of exposure and possible 226 use as a target for cancer diagnosis, prognosis and treatment. For example, the hub genes 227 identified in this study” - Please add a table for the hub genes that identified.

I recommend that the authors add a flowchart that highlights the entire experimental design.

Author Response

(The authors gave the same response as above.)

Round 2

Reviewer 2 Report

The revised version of the manuscript reports several improvements. The authors have well commented on all points of the review, making a set of changes that make the article comprehensive and mature. In my opinion, the article should be accepted in present form, I don't see any further changes needed.

Reviewer 3 Report

Accept in present form